# Ultrasensitive Surface Plasmon Resonance Sensor with a Feature of Dynamically Tunable Sensitivity and High Figure of Merit for Cancer Detection

**DOI:** 10.3390/s23125590

**Published:** 2023-06-14

**Authors:** Ravi Gollapalli, Jonathan Phillips, Puneet Paul

**Affiliations:** Department of Engineering and Industrial Professions, University of North Alabama, Florence, AL 35632, USA

**Keywords:** cancer detection, surface plasmon resonance, biosensor sensitivity, electric field, graphene, black phosphorus

## Abstract

Cancer is one of the leading causes of death worldwide, and it is well known that an early detection of cancer in a human body will provide an opportunity to cure the cancer. Early detection of cancer depends on the sensitivity of the measuring device and method, where the lowest detectable concentration of the cancerous cell in a test sample becomes a matter of high importance. Recently, Surface Plasmon Resonance (SPR) has proven to be a promising method to detect cancerous cells. The SPR method is based on the detection of changes in refractive indices of samples under testing and the sensitivity of such a SPR based sensor is related to the smallest detectable change in the refractive index of the sample. There exist many techniques where different combinations of metals, metal alloys and different configurations have been shown to lead to high sensitivities of the SPR sensors. Based on the difference in the refractive index between a normal healthy cell and a cancerous cell, recently, SPR method has been shown to be applicable to detect different types of cancers. In this work, we propose a new sensor surface configuration that comprises of gold-silver-graphene-black phosphorus to detect different cancerous cells based on the SPR method. Additionally, recently we proposed that the application of electric field across gold-graphene layers that form the SPR sensor surface can provide enhanced sensitivity than that is possible without the application of electrical bias. We utilized the same concept and numerically studied the impact of electrical bias across the gold-graphene layers combined with silver and black Phosphorus layers which forms the SPR sensor surface. Our numerical results have shown that electrical bias across the sensor surface in this new heterostructure can provide enhanced sensitivity compared to the original unbiased sensor surface. Not only that, our results have shown that as the electrical bias increases, the sensitivity increases up to a certain value and stabilizes at a still improved sensitivity value. Such dependence of sensitivity on the applied bias provides a dynamic tunability of the sensitivity and figure-of-merit (FOM) of the sensor to detect different types of cancer. In this work, we used the proposed heterostructure to detect six different types of cancers: Basal, Hela, Jurkat, PC12, MDA-MB-231, and MCF-7. Comparing our results to work published recently, we were able to achieve an enhanced sensitivity ranging from 97.2 to 1851.4 (deg/RIU) and FOM values ranging from 62.13 to 89.81 far above the values presented recently by other researchers.

## 1. Introduction

The World Health Organization has declared that cancer is one of the leading causes of death, responsible for approximately 10 million total deaths around the world. However, a significant number of these deaths could be avoided by early detection and related treatments. There are many methods and techniques to detect the different types of cancer by the identification of pre-cancer indicators prior to the development of cancer or sometimes before the symptoms exhibit themselves. Early detection of cancer not only results in a saved life, but has an economical benefit to it, as this will eliminate the need for expensive treatments [1]. With the need for this early intervention, many methods and technological solutions have been proposed and developed in an effort to not only efficiently and accurately detect cancer, but also to detect it years before the conventional diagnosis. Some of these recent methods are blood test [2], peptide-based optical biosensors [3], single extracellular vesicle analysis [4], serum protein profiling [5], FTIR spectroscopy for breast cancer detection [6], electroactive material-based immunosensors [7], photoelectrochemical sensors for pancreatic cancer [8], THz metasurface with array rectangular slot for skin detection [9], dual-signal amplified electrochemical biosensor for lung cancer detection [10], biomarkers and the microbiome for non-small cell lung cancer [11], prostate-specific antigen testing for prostate cancer detection [12], and two-step blood biomarker and PET imaging for early cancer detection [13]. Readers are directed to look to [14,15] for more information.

Among these various methods, recent developments have demonstrated that Surface Plasmon Resonance (SPR) is effective in both early detection and identification of different types of cancers [16,17,18]. The detection and identification process by an SPR sensor relies on the change in the refractive index (RI) of the sensing medium. Many methods and techniques to build SPR-based biosensors have been proposed by various researchers, along with techniques to enhance the sensitivity of these biosensors employing various combinations and configurations of sensor surfaces. In this report, we present a heterostructure consisting of gold-graphene-silver-black P as the sensor surface where the gold-graphene layer is electrically biased to tune the refractive index of the graphene layers to facilitate sensitivity enhancement and also provide dynamic tunability of the sensitivity for the detection of different types of cancers of cancers, such as Basal cell (skin cancer), HeLa (cervical cancer), Jurkat (blood cancer), PC12 (adrenal gland), MDA-MB-231 (breast cancer) and MCF-7 (breast cancer).

In the following sections we discuss the SPR sensor design, the modeling of electric bias across the gold-graphene layers, the numerical calculations of SPR curves, and related parameters for the heterostructures that are under consideration, along with the results, our discussion of the results and the conclusion.

## 2. Basic SPR System

A surface plasmon (SP) is a coherent electron oscillation that propagates together with an electromagnetic wave along a metal-dielectric interface. The surface plasmon can be excited, prominently, by two configurations, namely the Otto [19] and Kretschmann configuration [20]. In this report, we used the Kretschmann configuration as shown in Figure 1.

A basic SPR sensor consists of a sensor surface which is a thin metal film that interacts with bio-material, usually termed sensing medium. A change in the biomolecule concentration of the sensing medium results in a respective modification of the refractive index near the sensor surface. This modification results in a corresponding alteration in the propagation constant of the excitation optical wave, producing a property that can be optically evaluated [21]. Within the parameters of visible and near-infrared (IR) wavelengths, plasmon-supporting materials are limited to metals. However, these metals also suffer from ohmic losses. For the surface plasmon to be adequately excited, the metal film typically needs to consist of either gold (Au), silver (Ag), nickel (Ni), platinum (Pt), or copper (Cu) and have a thickness in the range of 45–55 nm. Gold (Au) offers the most promising properties, including excellent optical traits, adequate chemical stability, and high resistance to oxidation and corrosion. However, gold is also the most expensive metal and exhibits a lower biomolecule absorption rate in comparison with silver (Ag) [22,23,24,25]. To make use of the advantages of Au and Ag, both of them have been used as reported in other works [26,27]. One of the promising alternatives to improve the SPR sensor sensitivity is to increase the biomolecule absorption rate of sensor surcace by using graphene (Gr), which has significantly improved the sensitivity of the original system [28,29]. For the experimental part, as shown in Figure 1, a *p*-polarized laser beam is set incident on the prism-gold interface and the magnitude of the reflected beam is measured as a function of the angle of incidence. The optical prism behaves as a channel carrying the light to excite the surface plasmon at the prism-gold junction. Surface plasmon are excited at a specific angle of incidence, termed the surface plasmon resonance angle ($\theta_{spr}$), where a certain resonance condition is satisfied. The relationship governing the excitation of the surface plasmon is given as [30]
(1)$$\theta_{spr}=sin^{-1}\left(\frac{1}{n_{p}}\sqrt{\frac{n_{m}^{2}n_{a}^{2}}{n_{m}^{2}+n_{a}^{2}}}\right)\;,$$
where $n_{p}$ is the refractive index for the prism, $n_{m}$ is the refractive index for the metal, and $n_{a}$ is the refractive index for the analyte (sensing) medium. The angle of surface plasmon excitation is observed as a dramatic drop in the reflectance as shown in Figure A1. It should be noted that the surface plasmon resonance angle changes as the refractive index of the sensing medium changes. Based on this property, we can use SPR-based sensors, as they are utilized for detecting shifts in the resonance angles for various materials or material with different refractive indices.

The ‘angular sensitivity’ of an SPR system is represented as the ratio of the change in the SPR angle, $\Delta \theta_{spr}$ for the resulting variation in the refractive index, Δn of the sensing medium and is given as [31]
(2)$$S_{n}=\frac{\Delta \theta_{spr}}{\Delta n}.$$

In addition to angular sensitivity, other benchmarks are used to determine additional performative aspects of an SPR-based system. These additional metrics include detection accuracy (DA), and figure of merit (FOM) and are defined as [30]
(3)$$DA=\frac{1}{FWHM}$$
(4)$$FOM=\frac{S_{n}}{FWHM}$$

## 3. Theory of Biasing the Metal-Gr System

Graphite can be reduced into a two-dimensional, one-atom-thick sheet where the carbon atoms are positioned in a hexagonal lattice structure creating a material known as graphene. The Dirac point of graphene is observed where the conduction and valence bands intersect, and it determines where the chemical potential for undoped samples is located. The six Dirac points within graphene can be adjusted by applying an electrical voltage across the material [32,33], resulting in the potential to control the shift of the chemical potential away from the Dirac point. This phenomenon results in the tunability of the refractive index of graphene. To utilize this for the case of an SPR sensor, the simplest case of graphene (Gr) layers deposited on a substrate such as gold (Au) film, silver (Ag), or $SiO_{2}$ [34], etc., would form the sensor surface. For such an Au-Gr system, graphene’s carrier concentration, $n_{g}$, is dependent on the applied voltage, $V_{g}$ which can be defined as [35]
(5)$$n_{g}=\frac{V_{g}\varepsilon_{0}\varepsilon_{r}}{qd_{sub}}$$
where $\varepsilon_{0}=8.85\times 10^{-12}$ F/m, $\varepsilon_{r}$, *q*, and $d_{sub}$ are the permittivity of a vacuum, the relative permittivity of the substrate, the electron charge, and the substrate thickness, respectively. Dependent on the carrier concentration of the system, the chemical potential $\mu_{c}$ can be computed by [34]
(6)$$\mu_{c}=\hbar v_{f}\sqrt{\pi n_{g}}$$
where *ℏ* and $v_{F}$ are the reduced Planck’s constant and the Fermi velocity, $v_{F}=9.5\times 10^{5}$ cm/s, respectively. The optical conductivity of graphene, σ is the sum of the intra-band electron-photon scattering, $\sigma_{intra}$, and the inter-band electron transition conductivity, $\sigma_{inter}$, all being a function of the radiation frequency, ω, and modeled as
(7)$$\sigma \left(\omega \right)=\sigma_{intra}\left(\omega \right)+\sigma_{inter}\left(\omega \right)$$

Then $\sigma_{intra}$ and $\sigma_{inter}$ can be computed with the Kubo formula [36]:(8)$$\sigma_{intra}\left(\omega \right)=i\frac{q^{2}}{\pi \hbar^{2}\left(\omega +i\tau^{-1}\right)}\left[\mu_{c}+2K_{B}T\times ln\left\{e^{\left(\frac{-\mu_{c}}{K_{B}T}\right)}+1\right\}\right]$$
(9)$$\sigma_{inter}\left(\omega \right)=i\frac{q^{2}}{4\pi \hbar }ln\left[\frac{2|\mu_{c}|-\hbar \left(\omega +i\tau^{-1}\right)}{2|\mu_{c}|+\hbar \left(\omega +i\tau^{-1}\right)}\right]$$
where $K_{B}$, *T*, and τ are the Boltzmann’s constant, the temperature, the momentum relaxation time at $\tau =\mu_{c}m_{u}/qv_{F}^{2}$, and, $m_{u}=10^{4}$ cm${}^{2}$/Vs, is the impurity-limited direct current mobility.

Graphene’s complex conductivity is computed by:(10)$$\sigma \left(\omega \right)=\sigma_{R}\left(\omega \right)+i\sigma_{I}\left(\omega \right)$$
where $\sigma_{R}\left(\omega \right)$ and $\sigma_{I}\left(\omega \right)$ are calculated by Equations (Equation 7)–(Equation 9), representing the real and imaginary conductivity of graphene and are defined as
(11)$$\sigma_{R}\left(\omega \right)=\frac{\tau^{-1}q^{2}}{\left(\omega^{2}+\tau^{-2}\right)\pi \hbar^{2}}\times \left[\mu_{c}+2k_{B}T\times ln\left\{e^{\left(\frac{-\mu_{c}}{k_{B}T}\right)}+1\right\}\right],$$
(12)$$\sigma_{I}\left(\omega \right)=\frac{\omega q^{2}}{\left(\omega^{2}+\tau^{-2}\right)\pi \hbar^{2}}\times \left[\mu_{c}+2k_{B}T\times ln\left\{e^{\left(\frac{-\mu_{c}}{k_{B}T}\right)}+1\right\}\right]+\frac{q^{2}}{4\pi \hbar }ln\left[\frac{2|\mu_{c}|-\hbar \left(\omega +i\tau^{-1}\right)}{2|\mu_{c}|+\hbar \left(\omega +i\tau^{-1}\right)}\right].$$

The thickness of a single graphene layer is 0.34 nm, and for a given number of graphene layers with thickness, the relation between the relative permittivity and conductivity of graphene can be expressed as
(13)$$\varepsilon_{Gr}=1+i\frac{\sigma }{\omega \epsilon_{0}d_{Gr}}$$

Therefore, $d_{Gr}$ the real $n_{Gr,R}$ and imaginary $n_{Gr,I}$ parts of the graphene refractive index can be calculated as
(14)$$n_{Gr,R}=\sqrt{\frac{\sqrt{{\left(\sigma_{I}-\omega \varepsilon_{0}d_{Gr}\right)}^{2}+\sigma_{R}^{2}}-\left(\sigma_{I}-\omega \varepsilon_{0}d_{Gr}\right)}{2\omega \varepsilon_{0}d_{Gr}}}$$
(15)$$n_{Gr,I}=\sqrt{\frac{\sqrt{{\left(\sigma_{I}-\omega \varepsilon_{0}d_{Gr}\right)}^{2}+\sigma_{R}^{2}}+\left(\sigma_{I}-\omega \varepsilon_{0}d_{Gr}\right)}{2\omega \varepsilon_{0}d_{Gr}}}$$

From Equations (Equation 5)–(Equation 15), it is evident that the application of electrical bias across the metal-graphene sensor surface presents the potential to tune the refractive index of the sensor surface in the SPR biosensor. In the Kretschmann configuration, the reflectance of the incident light for this sensor surface with applied electrical bias can be calculated using the N-layer model [21].

## 4. Proposed SPR Sensor Surface

The heterostructure sensor surface we considered in this report is an Au-Gr-Ag-black phosphorus sensing medium, and the sensing medium here is the human sample cell under testing for cancer. The SPR setup for this heterostructure is shown in Figure 2. In this study, we started with the basic Au-Gr structure to utilize the advantages of increased sensitivity of the ’electrically biased sensor surface’ feature as explained in Section 3. Then we considered the Ag as our next layer based on its properties as explained in Section 2. The various layers of this heterostructure and their refractive indices are shown in Table 1.

### 4.1. Optimization of Gr, Au and Ag layer thicknesses

We optimized the thicknesses of Au, Gr and Ag layers of the proposed heterostructure by studying their impact on the SPR angle shift, FWHM, and the Rmin of the SPR curves primarily for the case of Basal normal cell (sensing medium). Xu et al. have shown that increasing the number of graphene layers results in increased sensitivity, so we chose to use 12 graphene layers in our study [37]. This is primarily due to the reason that we plan to build this SPR sensor, and one of the easy methods to deposit graphene layers is to use “graphene transfer sheets” from ACS Material, Pasadena, CA, USA. One of these graphene transfer sheets comes pre-packaged such that one can transfer between 6 and 8 graphene layers in a single deposition process. Therefore, by using two of such transfer sheets, we can deposit a minimum of 12 graphene layers, so we chose 12 graphene layers as optimal for our study [38]. Once the number of graphene layers was chosen, we optimized the number of Au-Ag layer thicknesses by changing the Au layer thickness from 25 nm to 40 nm, and the Ag layer thickness from 15 to 25 nm, while keeping the combined thickness of Au-Ag layers to be either 50 or 55 nm. We chose this number for the combined thickness because in a basic SPR system, the thickness of metal film is usually around 45-55 nm. From Figure 3, we see how the FWHM, Rmin, and SPR angle values vary for different Au-Ag layer thickness combinations. It should be noted that the best performance of an SPR system is obtained when it has the lowest FWHM, lowest Rmin, and the highest SPR angle shift for the smallest change in the sample refractive index, and considering these, it obvious that we can choose either Au:40-Ag:15 or Au:35-Ag:15. We chose Au:40-Ag:15 for our study as both have similar characteristics.

### 4.2. Optimization of Black Phosphorus Layers

With the successful integration of graphene in SPR sensors and the increased sensitivity it provides, the two-dimensional nanolayered materials have received tremendous interest in the scientific world. Among these 2D materials are graphene, black phosphorous (black P), $BaTiO_{3}$, $Ti_{3}C_{2}T_{x}\;MXene$, $MoS_{2}$, $Blue\;P/MoS_{2}$, and others have been studied by numerous researchers [39,40,41,42]. Among these, black P is the most stable among the allotropes of phosphorous atoms and also has structure similar to graphite. Multilayer black P has many unique features, such as the anisotropic electronic conductance, large carrier mobility, different optical responses, and the layer dependent electronic structures [43]. Because of these advantages, we chose to use black P as the functional layer that will be interacting with the sensing medium. To optimize the number of black P layers, we studied its impact on the FWHM, Rmin, and the SPR angles. We optimized the number of black phosphorus layers by varying the number of layers from 1 to 9 layers in increments of 2 layers (i.e., 1, 3, 5, 7, and 9 black P layers) and calculating the FWHM, Rmin, and the SPR angle shifts for the case of Au: 40 nm; Gr: 12 layers; Ag: 15 nm and black P layers for the case of Basal normal cell with a refractive index of 1.360. Figure 4a&b, show the variation of the SPR angles with regard to the applied chemical potential, for different black phosphorus layers. The SPR angle shifts over a wide range as the chemical potential increases.

From Figure 4a it is clear that as the number of the black P layers increase, the SPR angle occurs at higher values. For example, for the case of 9 layers, the SPR angle starts at 84${}^{\circ }$ and settles down at 79${}^{\circ }$ at 10 eV chemical potential, which is an increase of 5${}^{\circ }$ in the SPR angle, this is a welcoming feature. From Figure 5a, Rmin (the reflectance value at SPR angle) seems to be very high compared to other values of black P layers, an undesirable feature. From Figure 6a, it is clear that the increase in number of black P layers also results in an increased FWHM of the SPR curve, which is less desirable. In an SPR system, it is desirable to have a high SPR angle shift, low Rmin value, and the lowest FWHM value, and based on these, it makes sense to choose the number of black phosphorus layers to combine all these characteristics. Therefore, we chose five layers of black phosphorus for this study. Readers should note that black phosphorus is very unstable and will oxidize very rapidly in air and therefore care should be exercised to limit exposure of this layer to air.

### 4.3. Optimization of Prism Material

The optical prism is the primary element of an SPR system, as it is the one that couples the p-polarized light to excite the surface plasmon at the metal film. We optimized the prism material by considering the following prism material: N-K5, N-BK7, MgF2, SF10, and SK11. From Figure 7a, the SPR angle is 77.15${}^{\circ }$ at 1 meV and 74.05${}^{\circ }$ at 10 eV, which reveals a total SPR angle shift of 3.10°. When the prism material is SF10, the SPR angle is 59.25${}^{\circ }$ at 1 meV and 57.81${}^{\circ }$ at 10 eV, which reveals a total SPR angle shift of 1.44${}^{\circ }$. As N-K5 provides the maximum SPR angle shift, we chose it as the prism material for this study. Figure 7b & c show the change of FWHM and the Rmin at different chemical potential values.

**Table 1 sensors-23-05590-t001:** Proposed heterostructure layers and their refractive indices at λ=632.8 nm.

Layers	Refractive Index $n_{c}=n+\mathrm{ik}$	Thickness
Prism (N-K5) [44]	1.52064 (Section A.1)	-
Gold [45]	0.18228 + 1j*3.3776	40 nm
Graphene [46]	2.7411 + 1j*1.4016	4.08 nm (12 layers)
Silver [47]	0.056206 + 1j*4.2776	15 nm
Black Phosphorus [48]	3.5 + 1j*0.01	2.65 nm (5 layers)
Sensing Medium (cancer cells)	Refer to Table 2	-

### 4.4. Sensing Medium: Cancerous Cells

In this report, we considered cancerous cells as our sensing medium. X. J. Liang et al. reported the development of an integrated biochip to measure the refractive index for a single living cell and used it to measure the refractive indices of HeLa, PC12, MDA-MB-231, MCF-7 and Jurkat cancerous cells [49]. This has provided the means to use refractive index-based sensors to detect and identify different types of cancerous cells. Yaroslavsky et al. have shown the refractive index of a Basal cancerous cell [50]. The refractive indices of a cell sample taken from a human body and tested for the various types of cancers are provided in Table 2. The different types of cancers that can be tested using the SPR method, which is a method based on change of refractive index, are skin cancer, cervical cancer, blood cancer, adrenal gland cancer, and breast cancer. Liquid biopsy uses samples of blood, cerebrospinal fluid, urine, sputum, etc., which has recently been used for cancer testing [51,52,53].

**Table 2 sensors-23-05590-t002:** Refractive index (RI) variation between a normal cell and a cancerous cell [16,49].

Cancer Type	Cell Type	Normal Cell RI	Cancer Affected Cell RI
Skin	Basal	1.360	1.380
Cervical	HeLa	1.368	1.392
Blood	Jurkat	1.376	1.390
Adrenal Gland	PC12	1.381	1.395
Breast	MDA-MB-231	1.385	1.399
Breast	MCF-7	1.387	1.401

## 5. Results and Discussion

We used MATLAB software to perform the SPR calculations for all cancer types and compared the values of SPR angle, Rmin, values and the FWHM of each SPR curve between a healthy cell to its corresponding cancerous cell for all cancer types. Figure 8a shows the variation of the FWHM (deg) of the SPR curve at different applied chemical potential (eV) for both normal and cancerous cells of Basal, Hela, Jurkat cancer types, while Figure 8 b shows the variation of the FWHM for PC12, MDA-MB-231 & MCF-7 cancer types. Similarly, Figure 9a&b and Figure 10a&b, provide the Rmin and SPR angle values respectively for all cancer types.

In Table 3, the SPR angles at each applied chemical potential are shown. The sensitivity is calculated as the ratio of the SPR angle shift to the change in the refractive index, where the reference SPR angle is always taken to be at 1 meV chemical potential. For example, the sensitivity of the sensor at 1.5 eV for the case of Basal cancerous cell is calculated as (83.84−77.15)/(1.380−1.360)=334.8. Please note that here we are using the unbiased SPR angle for the normal cell as the reference to calculate the increase in the SPR angle for the cancerous cell. This shows sensitivity enhancement at each applied chemical potential value. The same method of referencing the SPR angle of the normal cell at 1 meV is used to calculate the FOM values for all the different types of cancerous cells. Therefore, to emphasize the novelty of ’electric bias’ across the sensor surface, column #4 of Table 3 reveals the values of the sensitivity. At 1 meV chemical potential, the sensitivity of the proposed sensor surface to the Basal cancerous cell is 237.6 as the refractive index changes from 1.360 (Basal normal cell) to 1.380 (Basal cancerous cell). As the chemical potential is increased to 1.5 eV, the sensitivity has changed to 334.8, an increase of 97.2 deg/RIU which goes to show that this method of applying electric bias results in an enhanced sensitivity of the SPR sensor. Similar argument can be made for the consideration of FOM values. This should be noted that this is an ’added’ sensitivity and ’added’ FOM of the SPR system, which is achieved without the need to make any changes to the sensor surface, such as changing the thickness of any of the Au, Gr, Ag or black P materials, and that the sensitivity can be tuned by choosing the appropriate value of chemical potential to achieve the maximum sensitivity or maximum FOM. This is the novelty and interesting characteristic of this method. The need of using FOM is clearly shown here, by looking at the SPR values in the table. For the case of HeLa cancerous cell at 1.0 eV, $S_{n}=340.50$, and, FOM=67.08, however, at 2.0 eV, $S_{n}=336$, and, FOM=71.79 which means that the FWHM at 2.0 eV is less than at 1.0 eV (Please refer to the definition of FOM shown in Equation (Equation 4)). Therefore, when we evaluate the performance of a SPR system, care should be given to both SPR angle shift as well as the FOM of the SPR curve. Researchers should also note that as the chemical potential is increased to higher values, the amount of ’added sensitivity’ or the additional gain in FOM is not prominent as compared to lower values of chemical potential, which means that to achieve high sensitivity, higher values of bias voltages are not needed, but rather lower values should be sufficient to see this effect in the real world. Table 4 clearly shows the amount of increased (maximum value) sensitivity and FOM for different types of cancers. Please note that the first column in this table is for the cancerous cell referenced to its corresponding normal cell. In Table 5, we compare the sensitivity and the FOM of other types of sensor surfaces reported recently. The proposed heterostructure with applied bias in this study results in very high values of sensitivity and FOM.

In Table 5, we compare the sensitivity and the FOM of our proposed heterostructure sensor surface to some recently reported work.

## 6. Conclusions

In this report, we present a heterostructure consisting of Au-Gr-Ag-black P as the SPR sensor surface to detect five different types of cancer (Basal, HeLa, Jurkat, PC12, MDA-MB-231 and MCF-7). The thicknesses of each layer of the sensor is optimized to yield high sensitivity and FOM; we achieved a maximum sensitivity of 1851.4 (deg/RIU) for the case of MCF-7 cancerous cell with a corresponding FOM of 63.80 (1/RIU) and a maximum sensitivity of 329.1 (deg/RIU) and corresponding FOM of 89.81 (1/RIU) for the case of Jurkat cancer cell. We have shown that when using electrical bias across the Au-Gr-Ag layers, the SPR angle of the sensor can be increased which results in a sensor that may provide better detection of different types of cancerous cells than provided by other methods. We have also shown that since the amount of shift in the SPR angle is dependent on the electrical bias voltage across the sensor, this technique also provides a handle to dynamically tune the sensitivity of such sensor.

## Figures and Tables

**Figure 1 sensors-23-05590-f001:**
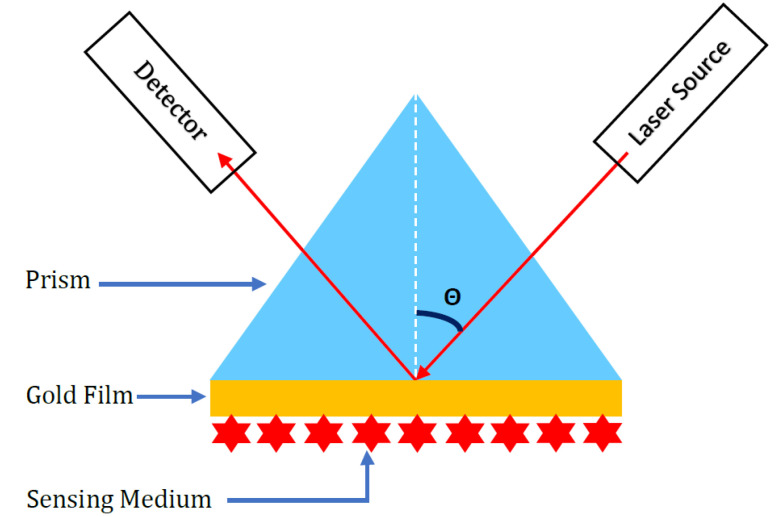
Kretschmann configuration with gold as the metal in the simple SPR system.

**Figure 2 sensors-23-05590-f002:**
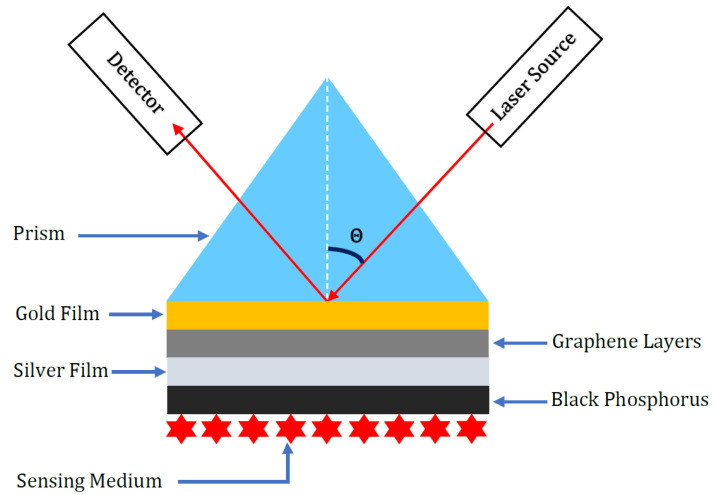
Kretschmann configuration showing prism (N-K5 material)-gold-graphene-silver-black phosphorus.

**Figure 3 sensors-23-05590-f003:**
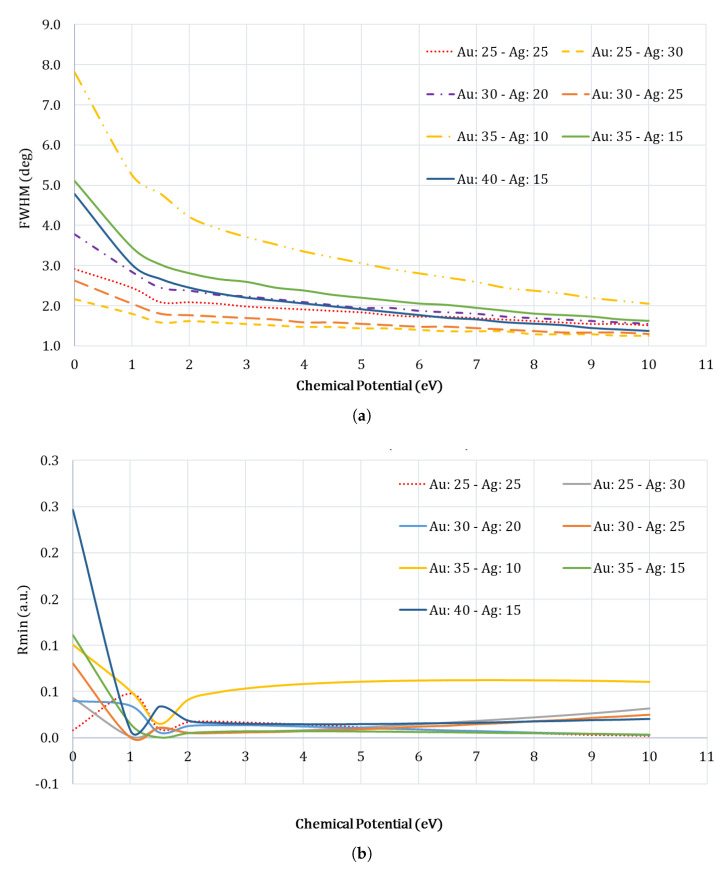
Au-Ag metal layer thicknesses were optimized by studying the FWHM, Rmin, and the SPR values at different chemical potentials, which was varied from 0 to 10 eV, for the case of Basal Normal Cell ($n_{Basal-normal-cell}=1.360$). (**a**) FWHM curves for different Au-Ag combinations; (**b**) Rmin values for different Au-Ag combinations; (**c**) SPR values for different Au-Ag combinations.

**Figure 4 sensors-23-05590-f004:**
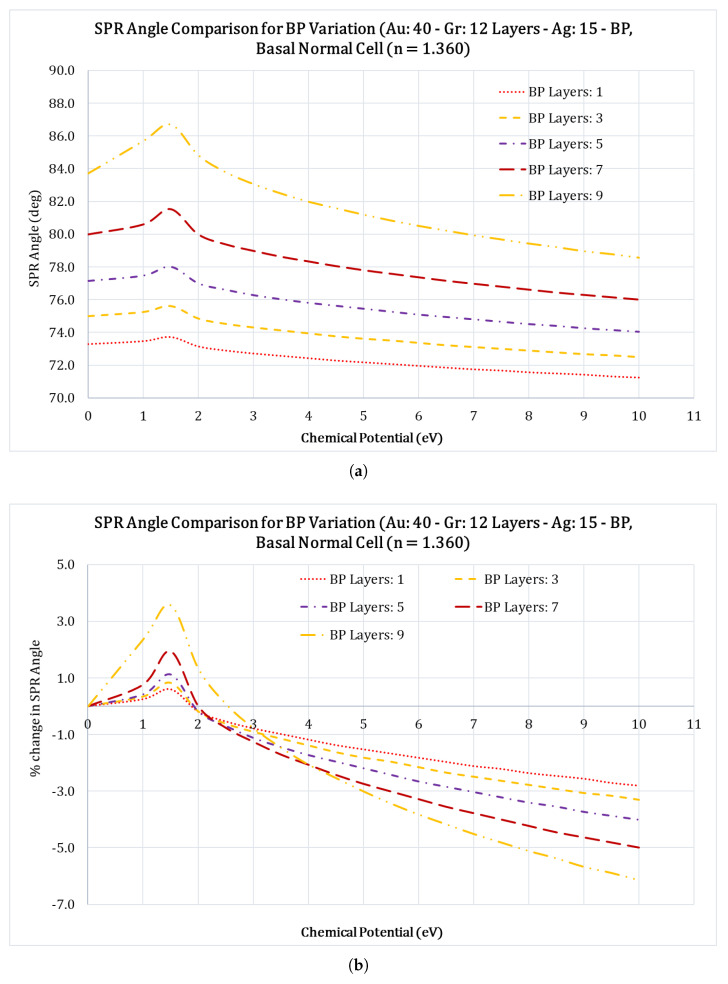
Comparison of the SPR angles vs. chemical potential (eV) for the case of a Basal normal cell ($n_{Basal-normal-cell}=1.360$) for different black P layers for the configuration Au: 40 nm; Gr: 12 layers; Ag: 15 nm; black P and Basal normal cell. **a** Variation of the SPR angle at different chemical potential values for different black P layers; **b** Percentage (%) change in the SPR angles at different chemical potential values for different black P layers.

**Figure 5 sensors-23-05590-f005:**
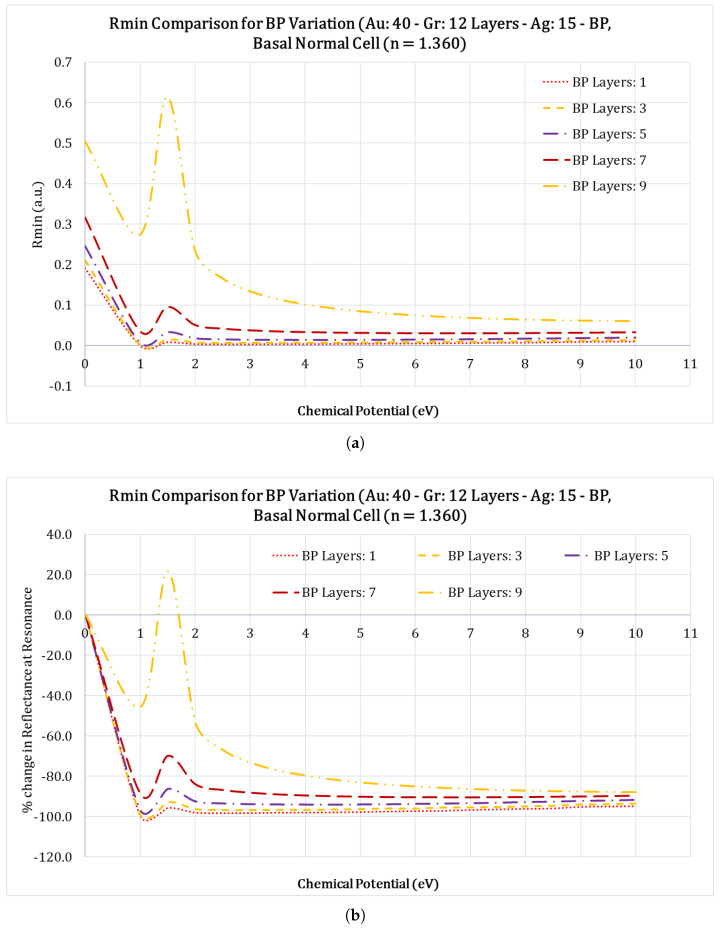
Figure shows the comparison of the Rmin vs. chemical potential (eV) for the case of a Basal normal cell ($n_{Basal-normal-cell}=1.360$) for different black P layers for the configuration Au: 40nm; Gr: 12 layers; Ag: 15 nm; black P and Basal normal cell. (**a**) Figure shows the variation of the Reflectance minimum value (Rmin) at different chemical potential values for different black P layers; (**b**) figure shows percentage (%) change in the Rmin at different chemical potential values for different black P layers.

**Figure 6 sensors-23-05590-f006:**
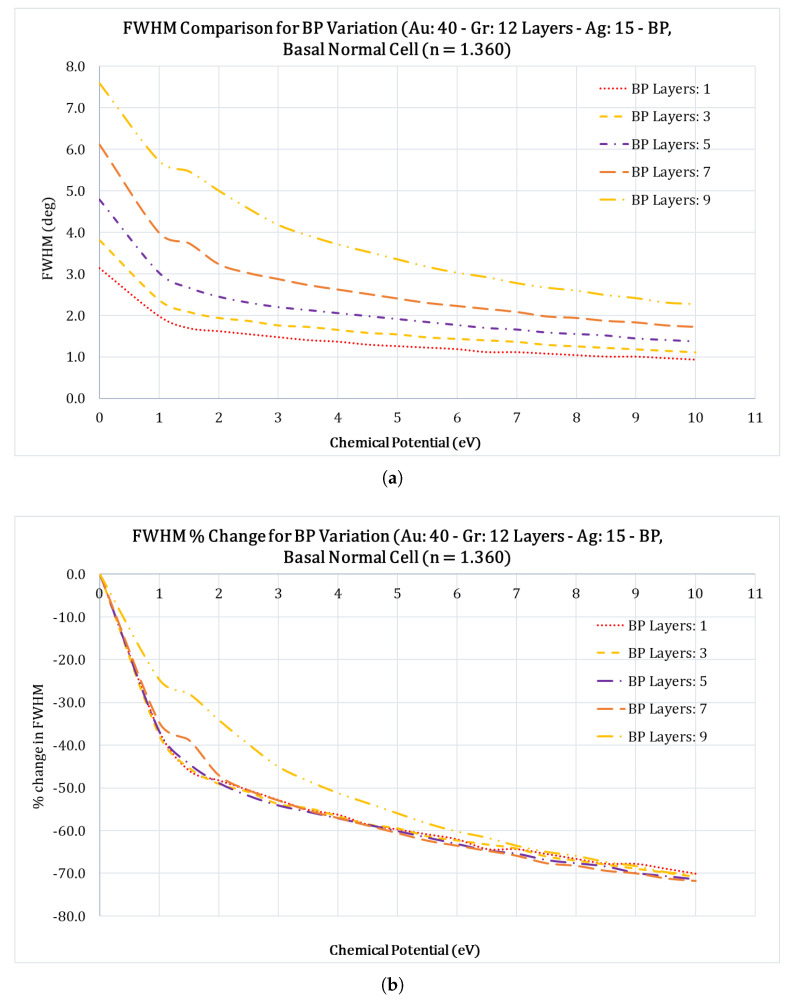
Comparison of the FWHM vs. chemical potential (eV) for the case of a Basal normal cell ($n_{Basal-normal-cell}=1.360$) for different black P layers for the configuration Au: 40nm; Gr: 12 layers; Ag: 15 nm; black P and Basal normal cell. (**a**) Variation of the FWHM at different chemical potential values for different black P layers; (**b**) Percentage (%) change in the Rmin at different chemical potential values for different black P layers.

**Figure 7 sensors-23-05590-f007:**
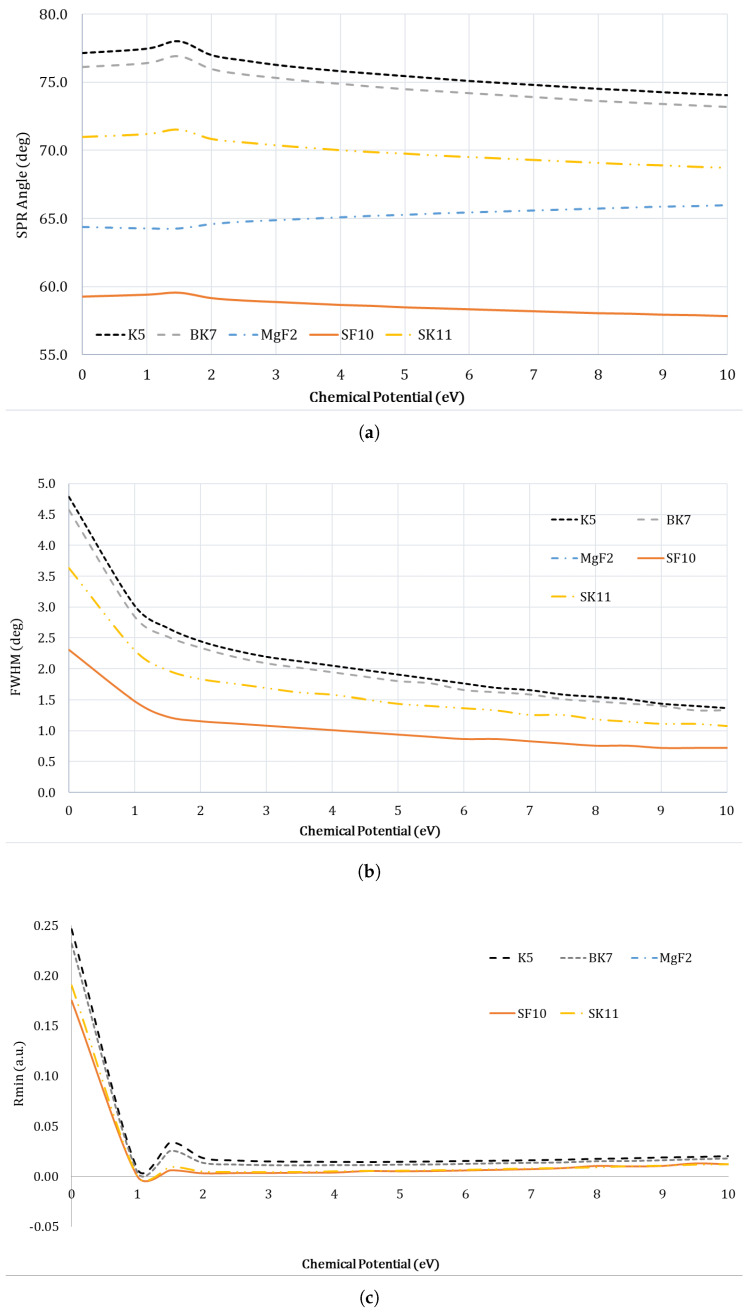
Prism material was optimized by considering the SPR angles, Rmin, and the FWHM of the curves as the chemical potential was varied from 0–10 eV, for the case of Basal normal cell ($n_{Basal-normal-cell}=1.360$). (**a**) SPR curves for different prism material; (**b**) FWHM for different prism material; (**c**) Rmin for different prism material.

**Figure 8 sensors-23-05590-f008:**
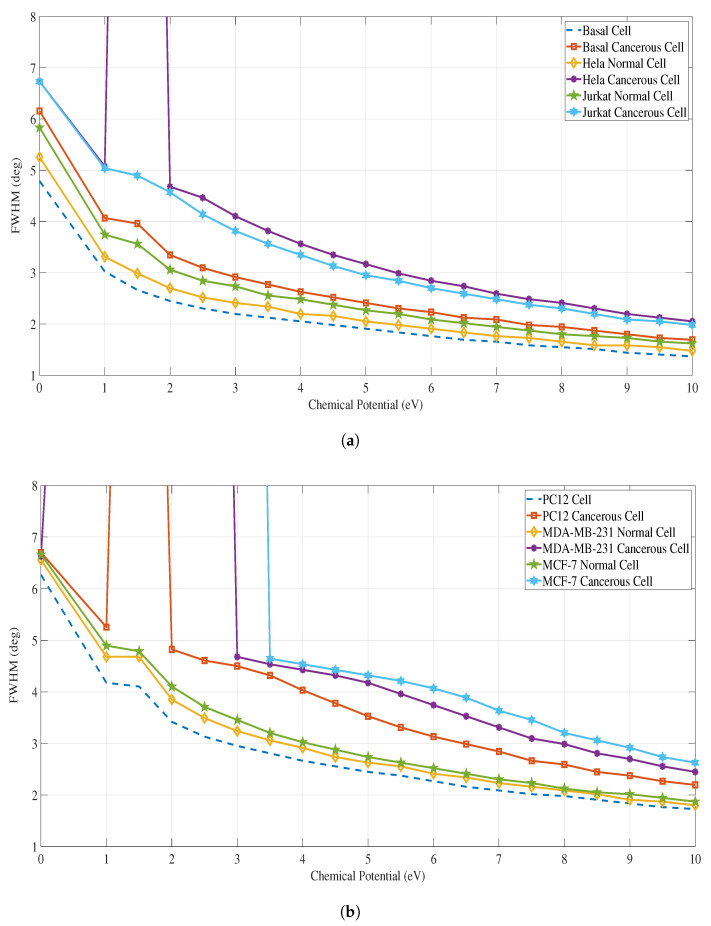
FWHM (deg) vs. chemical potential (eV) for each case of a normal cell and cancerous cell for all the types of cancer. Please refer to Table 2 for the refractive indices of these cells. (**a**) FWHM for Basal, HeLa and Jurkat normal and cancerous cells; (**b**) FWHM for PC12, MDA-MB-231, and MCF-7 normal and cancerous cells.

**Figure 9 sensors-23-05590-f009:**
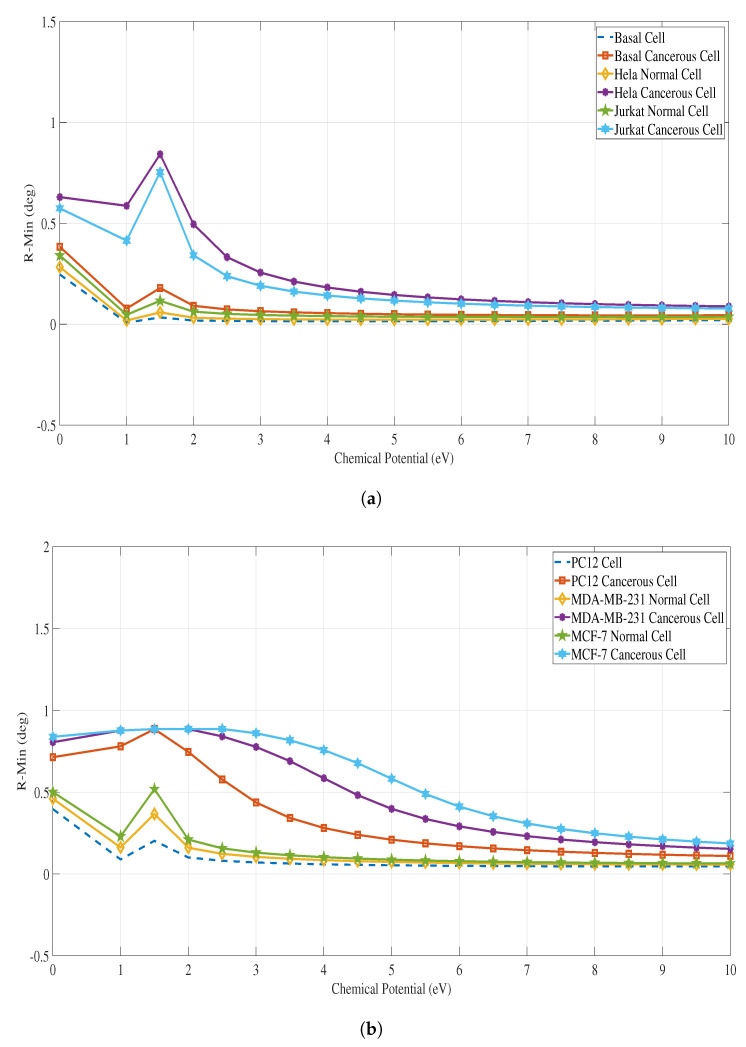
Rmin vs. chemical potential (eV) for each case of a normal cell and cancerous cell for all the types of cancer. Please refer to Table 2 for the refractive indices of these cells. (**a**) Rmin for Basal, HeLa and Jurkat normal and cancerous cells; (**b**) Rmin for PC12, MDA-MB-231, and MCF-7 normal and cancerous cells.

**Figure 10 sensors-23-05590-f010:**
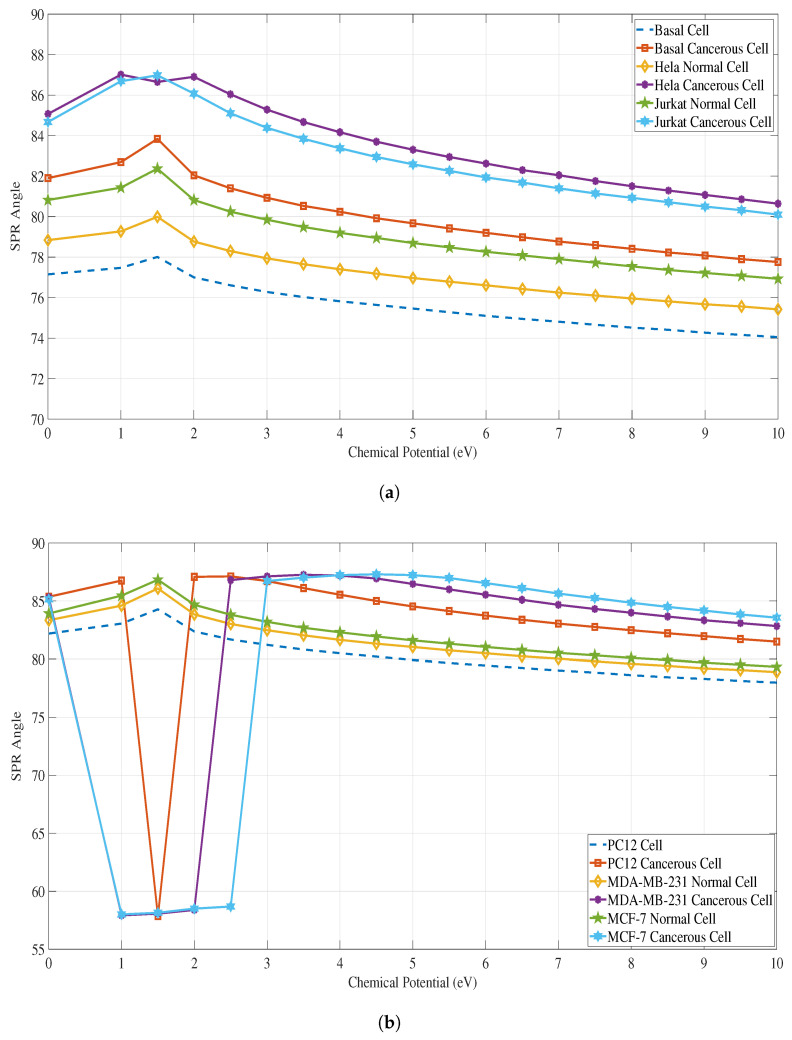
SPR angles (deg) vs. chemical potential (eV) for each case of a normal cell and cancerous cell for all the types of cancer. Please refer to Table 2 for the refractive indices of these cells. (**a**) SPR angles for Basal, HeLa and Jurkat normal and cancerous cells; (**b**) SPR angles for PC12, MDA-MB-231, and MCF-7 normal and cancerous cells.

**Table 3 sensors-23-05590-t003:** Table shows the variation of SPR angles (deg) wrt chemical potential (eV), the resulting sensitivity, $S_{n}$ and FOM. This table shows the values for the Basal (skin) normal and cancerous cells, along with the case of HeLa (cervical) normal and cancerous cells.

	Basal Normal Cell, *n* = 1.360	Basal Cancerous Cell, *n* = 1.380			HeLa Normal Cell, *n* = 1.368	HeLa Cancerous Cell, *n* = 1.392		
**$\mu_{c}$ (eV) ↓**	**$\theta_{\mathrm{spr}}$ (deg)**	**$\theta_{\mathrm{spr}}$ (deg)**	**$S_{n}$ wrt 1.360 (deg/RIU)**	**FOM wrt 1.360**	**$\theta_{\mathrm{spr}}$ (deg)**	**$\theta_{\mathrm{spr}}$ (deg)**	**$S_{n}$ wrt 1.368 (deg/RIU)**	**FOM wrt 1.368**
0.001	77.15	81.90	237.6	38.60	78.84	85.07	259.50	38.55
1.000	77.47	82.69	277.2	68.14	79.27	87.01	340.50	67.08
1.500	78.01	83.84	334.8	** 84.55 **	79.99	86.65	325.50	8.74
2.000	77.00	82.04	244.8	73.12	78.77	86.90	336.00	** 71.79 **
2.500	76.61	81.40	212.4	68.60	78.30	86.04	300.00	67.20
3.000	76.28	80.93	189.0	64.81	77.94	85.28	268.50	65.42
3.500	76.03	80.53	169.2	61.04	77.65	84.67	243.00	63.68
4.000	75.82	80.24	154.8	58.90	77.40	84.17	222.00	62.29
4.500	75.64	79.92	138.6	55.00	77.18	83.70	202.50	60.48
5.000	75.46	79.67	126.0	52.24	76.97	83.30	186.00	58.71
5.500	75.28	79.42	113.4	49.22	76.79	82.94	171.00	57.23
6.000	75.10	79.20	102.6	45.97	76.61	82.62	157.50	55.38
6.500	74.95	78.98	91.8	43.22	76.43	82.30	144.00	52.63
7.000	74.81	78.77	81.0	38.79	76.25	82.04	133.50	51.50
7.500	74.66	78.59	72.0	36.36	76.10	81.76	121.50	48.91
8.000	74.52	78.41	63.0	32.41	75.96	81.50	111.00	46.02
8.500	74.41	78.23	54.0	28.85	75.82	81.29	102.00	44.27
9.000	74.27	78.08	46.8	26.00	75.67	81.07	93.00	42.35
9.500	74.16	77.90	37.8	21.88	75.56	80.86	84.00	39.55
10.000	74.05	77.76	30.6	18.09	75.42	80.64	75.00	36.55

**Table 4 sensors-23-05590-t004:** Performance parameters of the proposed heterostructure for the six different types of cancer cells.

Cancer Cell Type	RI Change	SPR Angle Shift ${(}^{\circ })$	Max. SPR Angle Shift ${(}^{\circ })$ Due to Applied Bias with Optimal FOM	Sensitivity ${(}^{\circ }/\mathrm{RIU})$	Increased Sensitivity Due to Applied Bias	FOM (1/RIU)	Additional FOM Due to Applied Bias	Total FOM (1/RIU)
Basal	0.020	4.17	6.69	334.8	97.2	38.6	45.9	84.50
HeLa	0.024	6.23	8.06	336.0	76.5	38.5	33.2	71.74
Jurkat	0.014	3.85	6.16	329.1	54.0	40.8	48.9	89.81
PC12	0.014	3.17	4.93	388.2	162.0	33.7	42.6	76.45
MDA-MB-231	0.014	1.98	3.85	275.1	133.7	21.3	40.7	62.13
MCF-7	0.014	1.22	27.14	87.4	1851.4	2.2	61.5	63.80

**Table 5 sensors-23-05590-t005:** Comparison of the proposed work to some recently reported works.

Sensor Surace	Sensitivity ${(}^{\circ }/\mathrm{RIU})$	FOM (1/RIU)	Reported Year
$SF_{11}/Au/MoS_{2}/graphene$ [54]	130	17.02	2020
$Prism/Ag/PtSe_{2}/WS_{2}$ [55]	194	17.64	2020
$BK_{7}/Au/GeS$ [56]	260	33.40	2022
$BK_{7}/TiO_{2}/Au/graphene$ [18]	292.8	48.02	2022
Our present work, N−K5/Au/Gr/Ag/BP	408 ^1^	62.13	–
Our present work, N−K5/Au/Gr/Ag/BP	383.1 ^2^	89.81	–

^1^ MDA-MB-231 cancer - lowest FOM - with reference to Table 4. ^2^ Jurkat cancer - highest FOM - with reference to Table 4.

## Data Availability

All numerical data available upon request to the corresponding author.

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
