# Peer review of "Ultrasensitive Surface Plasmon Resonance Sensor with a Feature of Dynamically Tunable Sensitivity and High Figure of Merit for Cancer Detection"

_sensors, 2023, doi:10.3390/s23125590_

Round 1

Reviewer 1 Report

This manuscript reports that the application of electric field across the Au-Ag-Graphene-Black P sensor surface of a SPR system can provide additional sensitivity to an existing SPR sensor with a handle for dynamic tunability of the figure-of-merit (FOM) of the sensor. The whole manuscript is not well-organized. Considering the issues below, I think the work is inappropriate for “Sensors” publication.

1.Ref [42] proposed a highly-sensitive SPR sensor by combining the graphene-aluminum (Al)-graphene sandwich-like structure with Kretschmann configuration, where the configuration is similar with but simpler than that proposed in this manuscript. The advantages of the proposed structure are not clear compared to Ref [42].

2.The meaning of many acronyms like QRS, ECGs, FTIR, etc. mentioned in this manuscript are not clear.

3. Data points in Fig.10(b) are too sparse.

4.The wavelength of the incident light can largely influence the SPR excitation, but there lacks relative description.

Reviewer 2 Report

The author has presented work on SPR biosensors. The work may be useful for the scientific community. But more points need to be addressed properly before publication.

1.     The title of the manuscript is very confusing. What is the meaning of a Dynamically Tunable SPR biosensor?

2.     How the author calculated the FWHM for the proposed sensor?

3.     What is the sensing medium? What will the sample be if a cancer cell is the sensing medium?

4.     The plot given in the manuscript is very confusing. How the author calculated the performance parameters with respect to chemical potential?

5.     The comparative table consists of the different types of work. Means Ref [57] explains the wavelength interrogation technique for the proposed work. How can one compare different technology?

6.     I think this is not experimental work. So author needs to give an error tolerance analysis for the proposed work.

7.     Black phosphorus is very unstable in the air. It is oxidized when it comes in contact with air. How can it be prevented?

8.     Given that this is a purely theoretical work, please comment on how to verify the correctness of this work experimentally. And how to perform the experiments.

Reviewer 3 Report

The paper is very good.

there is an error in the relation (9)-hbar must be replaced with hbar^2.

See the paper from Optical and Quantum Electronics (2022) 54:328

Plasmonics based gas sensor with photonic spin hall effect in broad terahertz frequency range under variable chemical potential of graphene (A. K. Sharma et al.)

Also, in line 163, SPR angle is 77.47 deg for 1meV but in Table 3 this angle is for 1eV.

Also, in line 83, vf is 10^6 cm/s but in the paper of Sharma is 9.5x10^5 m/s. The refractive index of graphene from Table 1 is for a chemical potential miuc = 0 eV??

The value of time tau from the relations (9) and (10) must to be given.

Application of Electric Bias to Enhance the Sensitivity of Graphene-Based Surface Plasmon Resonance Sensors (Ravi Paul Gollapalli,Tingyi Wei and Jeremy Reid) Chapter (2022) in Intech Also is important as the authors to include a general relation between the conductivity and corresponding refractive index of the graphene in the electric field.

there is an error in relations (14) and (15). These must to be in accord with relation between relative permitivity and conductivity: epsilonG = 1+i(sigma)/omega(epsilon0)(dGr) Also on the line 90, V-s must to be replaced with Vs

Table 1 the refractive index of graphene is 2.7411+1.4016i. But this value is not as in cited reference 43[Gollapalli, Opt. Lett. 2020, 45, 2862]. This value is in Song et al 2018 from Refractive index info. Please to update this new very important observation.

Table 3 there is an error "the reference SPR angle is always taken to be at 1 meV chemical potential". Thus, for 1.5eV, the sensitivity is not (83.84-77.15)/0.02. The correct answer is (83.84-78.01)/0.02.

Round 2

Reviewer 1 Report

The revised manuscript is improved. It is recommended to accept after minor revision.

1.The abstract is too short and simple.

2. It is recommended that the authors modify the lengthy sentences to make them concise and to-the-point. For example, line 4-7, 17-21, 22-35, 121-124, 160-162, 192-196.

Author Response

Respected reviewer,

Thank you for your kind comments and suggestions. Please refer to the attached file for the edits I have made to the manuscript based on your comments/suggestions.

Reviewer 2 Report

The author has properly addressed all the comments.

Manuscript may be considered for the publication.

Author Response

Respected reviewer,

Thank you for your kind comment and your support towards this manuscript.